# Document-level Relationship Extraction by Bidirectional Constraints of Beta Rules

**Yichun Liu**[1*], **Zizhong Zhu**[2*], **Xiaowang Zhang**[1†]
**Zhiyong Feng**[1], **Daoqi Chen**[1], **Yaxin Li**[1]
[1]College of Intelligence and Computing, Tianjin University
[2]School of New Media and Communication, Tianjin University

## Abstract

Document-level Relation Extraction (DocRE) intends to extract relationships from documents. Some works introduce logic constraints into DocRE, addressing the issues of opacity and weak logic in original DocRE models. However, they only focus on forward logic constraints and the rules mined in these works often suffer from pseudo rules with high standard-confidence but low support. In this paper, we proposes Bidirectional Constraints of Beta Rules(BCBR), a novel logic constraint framework. BCBR first introduces a new rule miner which model rules by beta contribtion. Then forward and reverse logic constraints are constructed based on beta rules. Finally, BCBR reconstruct rule consistency loss by bidirectional constraints to regulate the output of the DocRE model. Experiments show that BCBR outperforms original DocRE models on relation extraction performance ($\sim$2.7 F1) and logic consistency($\sim$3.1 Logic). Furthermore, BCBR consistently outperforms two other logic constraint frameworks. Our code is available at https://github.com/Louisliu1999/BCBR.

## 1 Introduction

In recent years, DocRE attracts significant attention from researchers, with its intention to distinguish the relations between entity pairs in the documents. It's not limited to sentence-level relation extraction (Zeng et al., 2014; Zhang et al., 2017; Han et al., 2018; Wang et al., 2021). It aims to uncover the dependencies between entities in different sentences of one document (Zhou et al., 2021; Ma et al., 2023). The challenges in DocRE mainly include two aspects: first, it's difficult to capture complex long-range dependencies between entity pairs in documents; second, it's prone to errors in logical reasoning due to lack of logic. To address the aforementioned challenges, the academic community has made a lot of efforts.

* These authors contributed equally.
† Corresponding authors.

Figure 1: A case of logic constraint DocRE. Different colors represent different entities. Solid lines represent the relationships predicted by a general DocRE model, while dotted lines represent the relationships predicted by rule-based constraints.

Based on the use of rules, we can classify previous DocRE works into two categories: plain DocRE models and logic constraint DocRE models. In plain DocRE models, attention is mainly given to learning more powerful implicit representations. These methods include models based on sequence encoders (Wang et al., 2019; Xu et al., 2021a; Zhou et al., 2021), and models based on graph encoders (Zeng et al., 2020; Christopoulou et al., 2019). Although these methods have achieved decent results, their inferences are non-transparent and lack logic, making them prone to errors in logical inference. Meanwhile, the combination of relation extraction frameworks and logical rules has alleviated the issues of low transparency and weak logic. As the Figure 1 shows, we can derive two relations $based\_in0\,(Porsche, Argentina)$ and $gpe0\,(Argentina, Argentine)$ from the document. When the rule is applied,we can predict the relation $based\_in0\text{-}x$ between $Porsche$ and $Argentina$. LogiRE(Ru et al., 2021) is the first work that introduces logical rules into

document-level relation extraction. It employed the Expectation-Maximization (EM) algorithm to iteratively update the rule generator and relation extractor, optimizing the results of relation extraction. However, rule generator and relation extractor are in isolation. The EM algorithm enables joint optimization of the two models, but it can still lead to suboptimal results. MILR(Fan et al., 2022) addresses the issue of suboptimality by jointly training the relation classification and rule consistency losses. But MILR utilizes confidence-based methods to mine rules, which can lead to pseudo rules with high standard-confidence but low support, affecting the effectiveness of relation extraction.

In the paper, we propose BCBR, a novel framework which assists relation extraction with the help of logical rules. BCBR models the bidirectional constraints of beta rules and optimizes relation extraction through rule consistency loss. (1) The prior rule set derived from the documents is different from the one extracted from the knowledge graph. Textual data has a relatively small volume, and the inter-document correlations are low, leading to sparsity in the prior rule set. Thus, general rule miners in previous works cause the prevalence of pseudo rules with high standard-confidence but low support. To tackle this problem, we utilize the beta distribution to model the rules and consider both their successful predictions and failures to filter the rules. (2) In addition, we discovered that the constraints between rule head and rule body are bidirectional, while previous methods often only considered the forward constraints from rule body to rule head. Therefore, we introduce reverse constraints from rule head to rule body. (3) Finally, based on above constraints, we reconstruct the rule consistency loss to enhance the performance of the original DocRE models. We summarize our contributions as follows:

- To our knowledge, we first propose a rule miner that utilizes the Beta distribution to model rules.

- We introduce reverse logic constraint to ensure that the output of DocRE models satisfies the necessity of rules.

- We model bidirectional logic constraints as reasonable probability patterns and turn them into rule consistency loss.

- Our experiments demonstrate that BCBR sur-

passes LogiRE and MILR in terms of both relation extraction performance and logic consistency.

## 2 Related Work

### 2.1 Rule Learning

Rule learning is the foundation of logic constraint DocRE. Rule learning is primarily applied in the field of knowledge graphs, but DocRE can draw on its ideas. Currently, rule learning methods can be divided into three types: symbol-based rule learning, embedding-based rule learning, and differentiable rule learning based on TensorLog (Cohen, 2016). Symbol-based rule learning aims to mine rule paths of high frequency on knowledge graphs more efficiently. Galárraga et al. proposes the open-world assumption and utilizes pruning strategy to mine rules. Meilicke et al. adopts a bottom-up rule generation approach. Embedding-based rule learning focuses on learning more powerful embeddings for entities and relations. Omran et al. calculates the similarity between rule head and rule body to select better rules. Zhang et al. iteratively learns embeddings and rules to generate the rule set. TensorLog-based methods transform the rule learning process into a differentiable process, allowing neural network models to generate rules. For example, Sadeghian et al.; Sadeghian et al. trains a rule miner by using bidirectional RNN model, and Xu et al. utilizes transformer model.

### 2.2 Document-level Relation Extraction

Previous works on DocRE can be divided into two categories: plain DocRE and logic constraint DocRE. Plain document-level relation extraction focuses on learning more powerful representations (Zheng et al., 2018; Nguyen and Verspoor, 2018). There are methods based on sequence models that introduce pre-trained models to generate better representations (Wang et al., 2019; Ye et al., 2020; Xu et al., 2021b), Zhou et al. sets an adaptive threshold and uses attention to guide context pooling. Ma et al. uses evidence information as a supervisory signal to guide attention modules. Graph-based methods model entities and relations as nodes and edges in a graph and use graph algorithms to generate better representations (Zeng et al., 2020; Wang et al., 2020).

However, previous works lack transparency and logic, making them prone to errors in logical inference. Currently, research on logic constraint

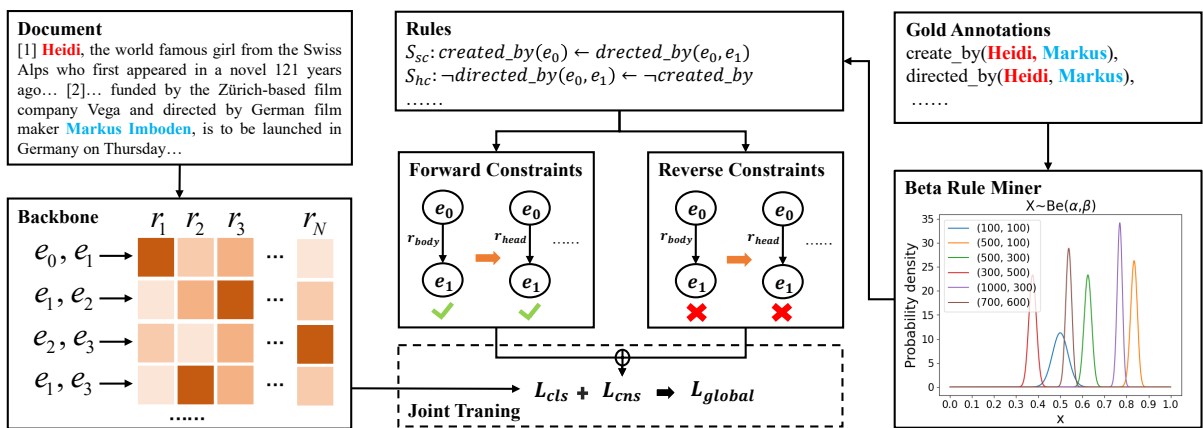

Figure 2: The framework of BCBR. The left is a plain DocRE model, the right is the Beta rule miner and the middle is the bidirectional logic constraint module and joint training module.

document-level relation extraction is limited. There are two noteworthy works in this area: LogiRE(Ru et al., 2021)and MILR (Fan et al., 2022). LogiRE involves two modules, the rule generator and the relation extractor. It uses the EM algorithm to efficiently maximize the overall likelihood. But this method often leads to suboptimal results due to the isolation between rule generator and relation extractor. To address this limitation, MILR constructs a joint training framework that combines rule consistency loss and relation classification loss of the backbone model. Previous works only applied forward logic constraints, while our works introduce reverse logic constraints and enhance the result of backbone model.

## 3 Method

In this chapter, we introduce our framework – Bidirectional Constraints of Beta Rules(BCBR) (Fig.2). We define concepts related to DocRE and rules(Sec.3.1). Then We propose a novel rule extraction method(Sec.3.2) and model bidirectional logic constraints based on rules(Sec.3.3). Finally, we construct rule consistency loss and jointly train with relation classification loss to enhance relation extraction performance(Sec.3.4).

### 3.1 Preliminaries

**Document-level Relation Extraction** Given a document $\mathcal{D}$ and entities $\mathcal{E} = \{e_i\}_1^n$. Entities constitute entity pairs $(e_h, e_t)_{1 \leq h,t \leq n, h \neq t}$, which $e_h$ and $e_t$ indicate the head entity and tail entity, respectively. The task of DocRE is to distinguish the relation $r$ between each entity pair $(e_h, e_t)$, where $r \in \mathbb{R}$ and $\mathbb{R} = \mathcal{R} \cup \{\mathcal{NA}\}$. $\mathcal{R}$ is a set of relations and $\mathcal{NA}$ indicates there is no relation on the entity pair.

**Logic Rules** We define a binary variable $r(e_h, e_t)$ to indicate the existence of $r \in \mathbb{R}$ between $e_h$ and $e_t$. When $r$ is true, $r(e_h, e_t) = 1$; otherwise, $r(e_h, e_t) = 0$. A rule consists of rule head and rule body. The rule head is denoted as $r_{head}(e_0, e_l)$, and the rule body is defined as the conjunction of $l$ binary variables, denoted as $r_{body}(e_0, e_l)$. We define the rule set as $\mathcal{S}$ and the pattern of rules is as follows:

$$r_{head}(e_0, e_l) \leftarrow r_1(e_0, e_1) \wedge ... \wedge r_l(e_{l-1}, e_l) \quad (1)$$

where $e_i \in \mathcal{E}$, $l$ represents the rule length, and $r_{head}(e_h, e_t)$ and $r_i(e_{i-1}, e_i)$ are referred to as the head atom and the body atom, respectively.

On the basis of Closed World Assumption(CWA)(Galárraga et al., 2015), we introduce two concepts: standard confidence and head coverage. Standard confidence refers to the conditional probability of rule head being satisfied given that the rule body is satisfied. The standard confidence of rule $s$ can be modeled as the following conditional probability distribution:

$$p_{sc}(s) = \frac{\mathcal{C}(r_{head} \wedge r_{body})}{\mathcal{C}(r_{body})} \quad (2)$$

$\mathcal{C}(\cdot)$ represents a counter.

Head coverage refers to the conditional probability of rule body being satisfied given that the rule head is satisfied. The head coverage of rule $s$ can be modeled as the following conditional probability distribution:

$$p_{hc}(s) = \frac{\mathcal{C}(r_{head} \wedge r_{body})}{\mathcal{C}(r_{head})} \quad (3)$$

**Backbone Model Paradigm** Our approach involves using logic rules to assist the original

DocRE model, which can be generalized to any backbone DocRE model. Therefore, we define the paradigm of the backbone model here. For all entity pairs $(e_h, e_t)$ in the document, the backbone model generates a score $\mathcal{G}(e_h, e_t)$ for their relation $r$. The probability of this triple being true is defined as follows:

$$P(r \mid e_h, e_t) = \sigma(\mathcal{G}(e_h, e_t)) \qquad (4)$$

where $\sigma(\cdot)$ is a sigmoid function.

During training, the backbone model uses binary cross-entropy loss or adaptive threshold loss to compute the relation classification loss $\mathcal{L}_{cls}$. During inference, the model sets a global threshold or uses a learned adaptive threshold to determine whether the triple $r(e_h, e_t)$ holds:

$$\rho_r(e_h, e_t) = \mathcal{I}(P(r \mid e_h, e_t) > \theta) \qquad (5)$$

where $\mathcal{I}(\cdot)$ represents the indicator function, and $\theta$ represents a threshold. If the probability of the triple being true satisfies the threshold, then $\rho_r(e_h, e_t) = 1$, indicating that the triple $r(e_h, e_t)$ holds. Conversely, if the probability does not satisfy the threshold, then $\rho_r(e_h, e_t) = 0$, indicating that the triple $r(e_h, e_t)$ does not hold.

## 3.2 Beta Rule Miner

Rule mining methods on knowledge graphs are mainly based on the large-scale and data-intensive essence of knowledge graphs. However, when these methods are transferred to document data (Ru et al., 2021; Fan et al., 2022), they still rely on confidence to filter rules. It leads to a inadaptable phenomenon that there are massive pseudo rules with high standard-confidence but low support. Therefore, we abandon the approach of using confidence or support alone and instead use the Beta distribution to model rules. In this section, we propose a new rule mining method called beta rule miner.

The Beta distribution $Beta(\alpha_s, \beta_s)$ for rule $s$ has two parameters, which we set as follows:

$$\alpha_s = \mathcal{C}(\varphi(s) = 1) + 1 \qquad (6a)$$
$$\beta_s = \mathcal{C}(\varphi(s) = 0) + 1 \qquad (6b)$$

where $\varphi(s)$ represents whether rule $s$ holds. Taking the example of mining high-confidence rules, if both $r_{body}$ and $r_{head}$ exist for rule $s$, then $\varphi(s) = 1$, indicating that the rule holds. Conversely, if $r_{body}$ exists for rule s but $r_{head}$ does not, then $\varphi(s) = 0$,

---

**Algorithm 1** Beta Rule Miner

**Input:** training set's labels : $\mathcal{T}$, rule template set generated by labels : $S_{template}$, lower bound of integration for Beta distribution : $k$, rule fitness threshold : $\eta$

**Output:** high quality rules : $S$
1: $S \leftarrow \{\}$
2: **for** $s$ in $S_{template}$ **do**
3: $\quad \alpha_s = 1$
4: $\quad \beta_s = 1$
5: $\quad$ **for** $\mathcal{T}_{\mathcal{D}}$ in $\mathcal{T}$ **do**
6: $\quad\quad$ **if** $r_{body}(e_0, e_l)$ in $\mathcal{T}_{\mathcal{D}}$ and $r_{head}(e_0, e_l)$ in $\mathcal{T}_{\mathcal{D}}$ **then**
7: $\quad\quad\quad \alpha_s \mathrel{+}= 1$
8: $\quad\quad$ **else if** $r_{body}(e_0, e_l)$ in $\mathcal{T}_{\mathcal{D}}$ and $r_{head}(e_0, e_l)$ not in $\mathcal{T}_{\mathcal{D}}$ **then**
9: $\quad\quad\quad \beta_s \mathrel{+}= 1$
10: $\quad\quad$ **end if**
11: $\quad$ **end for**
12: $\quad \rho_s = \Pi(P_s(x > k) > \eta)$
13: $\quad$ **if** $\rho_s == 1$ **then**
14: $\quad\quad S.add(s)$
15: $\quad$ **end if**
16: **end for**
17: **return** $S$

---

indicating that the rule does not hold. The probability density function of the Beta distribution for rule $s$ is given by:

$$f_s(x; \alpha_s, \beta_s) = \frac{x^{\alpha_s - 1}(1 - x)^{\beta_s - 1}}{\mathcal{B}(\alpha_s, \beta_s)} \qquad (7)$$

where $x \in [0, 1]$ and $\mathcal{B}(\cdot)$ represents the beta function. Next, we calculate the integration of the Beta distribution for rule $s$ (rule fitness). It determines whether rule $s$ is a high-quality rule or not.

$$P_s(x > k) = \int_k^1 f_s(x; \alpha_s, \beta_s) \, dx \qquad (8a)$$
$$\rho_s = \mathcal{I}(P_s(x > k) > \eta) \qquad (8b)$$

where $k$ is the lower bound of integration for Beta distribution and $\eta$ is the threshold for rule fitness. We select $s$ as a high-quality rule when the integration of its Beta distribution satisfies the threshold.

As shown in Algorithm 1, we summarize how to extract high quality rules. For each rule $s$, we calculates $\alpha_s$ and $\beta_s$ (lines 5-11). Then we computes $P_s(x > k)$ using equations (7) and (8a) (line 12) and add high standard-confidence rules to the rule set using equation (8b) (lines 13-17).

## 3.3 Bidirectional logic constraints

We utilize the above rules to impose constraints on the DocRE task. However, previous methods only employed forward logic constraints from $r_{body}$ to $r_{head}$(Fan et al., 2022). They could not leverage the reverse logic constraints from $r_{head}$ to $r_{body}$ due to the uncertainty of rule body atoms. BCBR models the reverse logic constraints based on head-coverage rules, thereby compensating for the loss of constraint conditions. Below, we provide a detailed explanation of the modeling process for bidirectional logic constraints:

**Forward logic constraints** Forward logic constraints exist in high standard-confidence rules. As shown in Equation (2), when $r_{body}$ occurs and $r_{head}$ simultaneously occurs, it is considered to satisfy the forward logic constraint. Conversely, if $r_{head}$ does not occur, it is considered not to satisfy the forward logic constraint. It represents the sufficiency of $r_{body}$ for $r_{head}$. We model the ideal form of forward constraints as follows:

$$P\left(r_{head}\left(e_0, e_l\right)\right) \geq b_{conf} * min\left(P\left(r_i\left(e_i, e_{i+1}\right)\right)\right), if\ min\left(P\left(r_i\left(e_i, e_{i+1}\right)\right)\right) > \theta \tag{9}$$

where $b_{conf}$ represents the rule fitness of high standard-confidence rules, $l$ denotes the length of the rule, and $\theta$ is a threshold. Forward constraints are only generated in high standard-confidence rules. When the score of the weakest body atom $r_i$ in $r_{body}$ is greater than $\theta$, the forward constraint of the rule comes into play. It constrains that the probability of $r_{head}$ being present is greater than the probability of $r_i$ being present.

**Reverse logic constraints** Reverse constraints exist in high head-coverage rules. As shown in the probability model in Equation (3), when $r_{head}$ is present, $r_{body}$ is also expected to be present. It is referred to as satisfying the reverse constraint. Conversely, if $r_{body}$ is not present, it is considered as not satisfying the reverse constraint. It represents the necessity of $r_{body}$ for $r_{head}$. The reverse constraint is formulated as shown in Equation (10a). Reverse constraint differs from the rule form of forward constraint shown in Equation (1), as it derives $r_{body}$ from $r_{head}$. $r_{body}$ contains multiple uncertain body atoms because the entities connecting the triple may not exist. But conjunction rules require to consider each triple in constructing the constraint probability model. So We use De Morgan's laws for (10a) and obtain a disjunctive rule as shown in

Equation (10b), which states that if any body atom does not exist, then $r_{head}$ does not exist.

$$r_{head}\left(e_0, e_l\right) \rightarrow r_1\left(e_0, e_1\right) \wedge ... \wedge r_l\left(e_{l-1}, e_l\right) \tag{10a}$$

$$\neg r_{head}\left(e_0, e_l\right) \leftarrow \neg r_1\left(e_0, e_1\right) \vee ... \vee \neg r_l\left(e_{l-1}, e_l\right) \tag{10b}$$

We model the ideal probability form of the reverse constraint as Equation (11):

$$P\left(r_{head}\left(e_0, e_l\right)\right) \leq b_{head} * min\left(P\left(r_i\left(e_i, e_{i+1}\right)\right)\right), if\ min\left(P\left(r_i\left(e_i, e_{i+1}\right)\right)\right) < \theta \tag{11}$$

where $b_{head}$ represents the rule fitness of high head-coverage rules. The inverse constraint is only generated in high head-coverage rules. When the score of weakest body atom $r_i$ in $r_{body}$ is less than $\theta$, the reverse constraint of the rule comes into effect. It constrains that the existence probability of $r_{head}$ must be less than the existence probability of $r_i$.

## 3.4 Rule consistency loss

In addition to the original relation classification loss $\mathcal{L}_{cls}$ of backbone models, we construct a rule consistency loss based on the bidirectional constraints of beta rules. This loss is jointly trained with $\mathcal{L}_{cls}$ to improve the logical consistency and performance of relation extraction.

The rule consistency loss is derived from the bidirectional constraints of beta rules and consists of two parts: forward loss generated by high standard-confidence rules and reverse loss generated by high head-coverage rules. The loss function is formulated as shown in equations (12a) and (12b).

$$\mathcal{L}_{sc} = \sum_{s \in S_{sc}} \sum_{d \in \mathcal{D}} max\left(0, \left(\log\left(b_{conf}\right) + \log\left(min\left(P\left(r_i \mid e_{i-1}, e_i\right)\right)\right) - \log\left(P\left(r_{head} \mid e_0, e_l\right)\right)\right)\right) * \rho_{r_{min}}\left(e_h, e_t\right) \tag{12a}$$

$$\mathcal{L}_{hc} = \sum_{s \in S_{sc}} \sum_{d \in \mathcal{D}} max\left(0, -\left(\log\left(b_{head}\right) - \log\left(min\left(P\left(r_i \mid e_{i-1}, e_i\right)\right)\right) + \log\left(P\left(r_{head} \mid e_0, e_l\right)\right)\right)\right) * \rho_{r_{min}}\left(e_h, e_t\right) \tag{12b}$$

where $S_{sc}$ and $S_{hc}$ represent the sets of high standard-confidence rules and high head-coverage rules, respectively. $b_{conf}$ and $b_{head}$ are their rule fitness. $\rho_{r_{min}}\left(e_h, e_t\right)$ is an indicator function mentioned in equation (5), which takes 1 if the weakest triple in the rule body holds true, and 0 otherwise.

We combine the bidirectional constraint losses

into a unified loss, which is jointly computed with $\mathcal{L}_{cls}$. The formulation is as follows:

$$\mathcal{L}_{global} = \mathcal{L}_{cls} + \lambda * (\mathcal{L}_{sc} + \mathcal{L}_{hc}) \quad (13)$$

where $\lambda$ is a relaxation factor that reflects the weight of the rule consistency loss.

## 4 Experiments

### 4.1 Datasets

To demonstrate the ability of our method to generalize, we conducted evaluations on three datasets for document-level relation extraction. including DWIE(Zaporojets et al., 2021), DocRED(Yao et al., 2019), and Re-DocRED(Tan et al., 2022). The detailed information of datasets are shown in Table 1.

**DWIE** This dataset is a human-annotated collection used for document-level information extraction, which includes DocRE. It contains gold rule labels, which can be used to evaluate the logical consistency of the output of DocRE models.

**DocRED** It's a popular large-scale DocRE dataset, which is sourced from Wikipedia articles. It is the most widely used dataset for DocRE, and the majority of methods are experimented on this dataset.

**Re-DocRED** It analyzes the causes and impacts of false negatives in the DocRED dataset and reannotates 4,053 documents. Compared to the DocRED dataset, most document-level relation extraction methods show significant improvement in performance on this dataset.

| Dataset | | #Doc. | #Rel. | Avg.#Ent. |
|---|---|---|---|---|
| | Train | 602 | | 27.4 |
| DWIE | Dev | 98 | 65 | 28.4 |
| | Test | 99 | | 26.5 |
| | Train | 3053 | | 19.5 |
| DocRED | Dev | 1000 | 96 | 19.6 |
| | Test | 1000 | | 19.5 |
| | Train | 3053 | | 19.4 |
| Re-DocRED | Dev | 500 | 96 | 19.4 |
| | Test | 500 | | 19.6 |

Table 1: Statistics of datasets.

### 4.2 Experimental Setups

**Metrics** Following the experimental settings of (Ru et al., 2021) and (Fan et al., 2022), we evaluate our method using three metrics: F1, Ign F1, and Logic. The Ign F1 score excludes relation triplets that involved by either train set or dev set, preventing leakage of information from the test set. Logic is used to assess the adherence of our predictions to the golden rule.

**Baselines** To verify the generalizability of our method as a plugin model for DocRE, we select the following four models as backbone models: BiL-STM(Yao et al., 2019), GAIN(Zeng et al., 2020), ATLOP(Zhou et al., 2021), and DREEAM(Ma et al., 2023). For fairness, we choose bert-base-cased as the pretraining model for GAIN, ATLOP, and DREEAM. Meanwhile, we also compare our model BCBR with other logic constraint DocRE models – LogiRE(Ru et al., 2021) and MILR(Fan et al., 2022)[1].

**Implementation Details** For fairness, we conduct experiments based on the recommended parameters in the baselines. We average the results over five different random seeds. The specific hyper-parameter settings for the new parameters introduced by BCBR are provided in Appendix A. All models were implemented using PyTorch 1.8.1 and trained on a Quadro RTX 6000 GPU.

### 4.3 Results & Discussions

**Results on DWIE** We can observe results on DWIE in Table 2. Among all baseline models, our BCBR model consistently outperforms LogiRE and MILR, indicating its strong generality, making it compatible with the majority of DocRE models. Building upon the state-of-the-art baseline model, DREEAM, BCBR achieves 3.33% Ign F1, 3.34% F1 and 4.02% Logic improvements on test set, reaching state-of-the-art performance. In comparison to LogiRE and MILR, BCBR achieves 1.94% Ign F1, 1.40% F1 and 2.83% Logic improvements. It demonstrates that BCBR has achieved significant improvements in both relation extraction performance and rule consistency. Meanwhile, We conducted a comparative experiment between our beta rule miner and a general rule miner, as detailed in the Appendix B.

**Results on DocRED** The experimental results on DocRED are presented in Table 2. Apart from DWIE, we only include the performance of strong baselines on other datasets. LogiRE does not exhibit significant improvements on the DocRED dataset, primarily due to the presence of a large number of false negative labels. The EM algorithm used in LogiRE leads to overfitting issues. On the other hand, MILR and BCBR perform rela-

---

[1]Our implementation.

| model | Dev | | | Test | | |
|---|---|---|---|---|---|---|
| | Ign F1 | F1 | Logic | Ign F1 | F1 | Logic |
| BiLSTM | 40.46 | 51.92 | 64.87 | 42.03 | 54.47 | 64.41 |
| BiLSTM+LogiRE | 42.59(+2.13) | 53.83(+1.91) | 73.37(+8.50) | 43.65(+1.62) | 55.14(+0.67) | 77.11(+12.70) |
| BiLSTM+MILR | 43.03(+2.57) | 53.90(+1.98) | 74.66(+9.79) | 43.80(+1.77) | 55.48(+1.01) | 77.69(+13.28) |
| BiLSTM+BCBR | 43.71(+3.25) | 54.61(+2.69) | 76.01(+11.14) | 45.46(+3.43) | 57.13(+2.66) | 79.85(+15.44) |
| GAIN | 58.63 | 62.55 | 78.30 | 62.37 | 67.57 | 86.19 |
| GAIN+LogiRE | 60.12(+1.49) | 63.91(+1.36) | 87.86(+9.56) | 64.43(+2.06) | 69.40(+1.83) | 91.22(+5.02) |
| GAIN+MILR | 60.44(+1.81) | 64.03(+1.48) | 83.59(+5.29) | 65.19(+2.82) | 70.17(+2.60) | 87.67(+1.48) |
| GAIN+BCBR | 61.37(+2.74) | 64.83(+2.28) | **88.29**(+9.99) | 66.72(+4.35) | 71.25(+3.68) | **91.69**(+5.40) |
| ATLOP | 59.03 | 64.82 | 81.98 | 62.09 | 69.94 | 82.76 |
| ATLOP+LogiRE | 60.24(+1.21) | 66.76(+1.94) | 86.98(+5.00) | 64.11(+2.02) | 71.78(+1.84) | 86.07(+3.31) |
| ATLOP+MILR | 59.58(+0.55) | 65.51(+0.69) | 86.32(+4.34) | 65.08(+2.99) | 71.85(+1.91) | 86.94(+4.18) |
| ATLOP+BCBR | 60.91(+1.88) | 66.44(+1.62) | 87.13(+5.45) | 66.25(+4.16) | 73.19(+3.25) | 90.27(+7.51) |
| DREEAM | 60.84 | 66.07 | 82.43 | 64.82 | 71.44 | 84.78 |
| DREEAM+LogiRE | 61.53(+0.69) | 66.84(+0.77) | 84.06(+1.63) | 65.79(+0.97) | 73.02(+1.58) | 85.27(+0.49) |
| DREEAM+MILR | 61.39(+0.55) | 66.51 (+0.44) | 83.49(+1.06) | 66.21(+1.39) | 73.38(+1.94) | 85.97(+1.19) |
| DREEAM+BCBR | **62.23**(+1.39) | **68.07**(+2.00) | 84.69(+2.26) | **68.15**(+3.33) | **74.78**(+3.34) | 86.07(+4.02) |

Table 2: Main results on DWIE(%).

| model | Test | |
|---|---|---|
| | Ign F1 | F1 |
| GAIN | 57.93 | 60.07 |
| GAIN+LogiRE | 58.62(+0.69) | 60.61(+0.54) |
| GAIN+MILR | 58.85(+0.92) | 61.01(+0.96) |
| GAIN+BCBR | 59.36(+1.43) | 61.37(+1.30) |
| ATLOP | 58.28 | 60.29 |
| ATLOP+LogiRE | 58.52(+0.24) | 60.41(+0.12) |
| ATLOP+MILR | 59.07(+0.79) | 60.98(+0.69) |
| ATLOP+BCBR | 59.89(+1.61) | 61.63(+1.44) |
| DREEAM | 59.08 | 60.86 |
| DREEAM+LogiRE | 59.29(+0.21) | 61.03(+0.17) |
| DREEAM+MILR | 60.13(+1.05) | 61.78(+0.92) |
| DREEAM+BCBR | **60.77**(+1.59) | **62.39**(+1.53) |

Table 3: Main results on DocRED(%).

| model | Test | |
|---|---|---|
| | Ign F1 | F1 |
| GAIN | 69.77 | 70.59 |
| GAIN+LogiRE | 70.53(+0.76) | 71.48(+0.89) |
| GAIN+MILR | 70.82(+1.05) | 71.78(+1.19) |
| GAIN+BCBR | 71.57(+1.80) | 72.34(+1.75) |
| ATLOP | 70.86 | 71.68 |
| ATLOP+LogiRE | 71.83(+0.97) | 72.77(+1.09) |
| ATLOP+MILR | 71.86(+1.00) | 72.58(+0.90) |
| ATLOP+BCBR | 72.43(+1.57) | 73.22(+1.54) |
| DREEAM | 71.45 | 72.16 |
| DREEAM+LogiRE | 72.23(+0.78) | 72.92(+0.76) |
| DREEAM+MILR | 72.28(+0.83) | 73.03(+0.87) |
| DREEAM+BCBR | **72.74**(+1.29) | **73.50**(+1.34) |

Table 4: Main results on Re-DocRED(%).

tively better as they jointly train with DocRE models. BCBR achieves the best results on this dataset, with 1.53% Ign F1 and 1.59% F1 improvements. It demonstrates that BCBR performs better than LogiRE and MILR on DocRED.

**Results on Re-DocRED**  The results on Re-DocRED can be seen from Table 4. Due to the resolution of false-negative labels in DocRED, most relation extraction models exhibit significant improvements on this dataset. BCBR achieves 1.34% Ign F1 and 1.29% F1 improvement, which is slightly higher than the improvement on DocRED. By this, we can conclude that the BCBR can assist the backbone model more effectively when a majority of the false-negative label issues are resolved.

## 4.4   Ablation study

To demonstrate the effectiveness of each component of the BCBR framework, we conduct ablation experiments, and the experimental results are shown in Table 5. We use the DWIE dataset and perform the experiments on the strongest baseline model DREEAM. In the table, BR and BC refer to the Beta Rule and Bidirectional Constraint, respectively. We exclude the beta rules using the original rule miner and exclude the bidirectional constraint using the rule forward constraint. From the table, we can observe that when exclude one of the components, our method still outperforms the baseline approach. This indicates that both components are effective. The quality of rules and the comprehensiveness of logic constraints are both

| | | |
|---|---|---|
| **Document** | [1] All season long, **Bayern** Munich have been consumed with achieving one goal: reaching the final of the Champions League…[2] After finishing a distant second place to **Dortmund** in the Bundesliga and dismally losing to them 5-2 in the German Cup final… | **Predictions of DREEAM:**
vs
**Bayern** → **Dortmund**

**Predictions of BCBR:**
vs
**Bayern** → **Dortmund**
vs |
| **Rule** | $s_{sc}$: $vs(e_0, e_1) \leftarrow vs(e_1, e_0)$ | |
| **ChatGPT** | **Prompt**: …if there is a relationship {vs} from Bayern to Dortmund , please choose the relationship from Dortmund to Bayern from the following relationships: {vs}, {won_vs}… None.
**Output**: {vs} | |
| **Documents** | [1] This is **Modi**'s fourth trip to the US after taking office as prime minister, and the **Indian** leader's schedule in the US capital includes holding talks with President Barack Obama...[2] …to boost US investment into **India**, particularly in the energy sector. | **Predictions of DREEAM:**
**Modi** → **Indian** → **India**
head_of_gov-x  gep0

**Predictions of BCBR:**
head_of_gov
**Modi** → **Indian** → **India**
head_of_gov-x  gep0 |
| **Rule** | $s_{sc}$: $head\_of\_gov(e_0, e_2) \leftarrow head\_of\_gov\text{-}x(e_0, e_1) \land gep0(e_1, e_2)$ | |
| **ChatGPT** | **Prompt**: … if there is a relationship {head_of_gov-x} from Modi to Indians, and a relationship {gep0} from Indian to India, please choose the relationship from Modi to India from the following relationships: {head_of_gov-x}, {head_of_gov}, {head_of_state}, {head_of_state-x}, … None.
**Output**: {head_of_gov-x} | |
| **Documents** | [1] **Berlin court** rules Google Street View is legal in **Germany**. [2]Last Tuesday, the **Berlin State Supreme Court** (Kammergericht) announced its… [3] allow **Germans** to opt-out of the service to have their house obfuscated as well. | **Predictions of DREEAM:**
based_in0  **Berlin court**  agency_of-x
**Germany** → **Germans**
gep0
**Predictions of BCBR:**
based_in0  **Berlin court**
**Germany** → **Germans**
gep0 |
| **Rule** | $s_{hc}$: $\neg agency\_of\text{-}x(e_0, e_2) \leftarrow \neg based\_in0(e_0, e_1) \lor \neg gep0(e_1, e_2)$ | |
| **ChatGPT** | **Prompt**: …if there is a relationship {gep0} from Germany to Germans, and not a {based_in0} from Berlin court to Germany please choose the relationship from Berlin court to Germans from the following relationships: {gep0}, {agency_of-x}, … None.
**Output**: {agency_of-x} | |

Figure 3: Several BCBR inference cases on DWIE and the predictions of the large language model-ChatGPT on them. Different colors represent different entities. Green solid lines represent correct predictions, red solid lines represent incorrect predictions, and gray dotted lines represent non-existent relations that correspond to rules.

crucial factors.

| model | Dev | | Test | |
|---|---|---|---|---|
| | Ign F1 | F1 | Ign F1 | F1 |
| DREEAM+BCBR | **62.23** | **68.07** | **68.15** | **74.78** |
| DREEAM+BC | 61.94 | 67.86 | 67.74 | 74.42 |
| DREEAM+BR | 60.94 | 67.19 | 66.57 | 73.55 |
| DREEAM | 60.83 | 66.07 | 64.82 | 71.44 |

Table 5: Ablation study on the DWIE dataset(%).

## 4.5 Case study & LLM

We list some rules mined by our Beta Rule Miner and their beta scores in Table 6. From the table, we can learn about various rule patterns that we can mine. Then we present several inference cases of BCBR framework on the DWIE dataset, as shown in Figure 3. We compare the results of BCBR with the strongest baseline - BREEAM and highlight the advantages of using logic constraints. We also compare ours to the outputs of large language models to demonstrate the significance of our task during the era of large language models. In the first case, BREEAM can only predict the relation $vs$ from $Bayern$ to $Dortmund$, while our BCBR can predict the relation $vs$ from $Dortmund$ to $Bayern$ due to the assistance of rules. However, ChatGPT

also choose the correct answer, which indicates that general DocRE models cannot even perform simple logical inference, but there is no significant gap between our framework and ChatGPT in simple logical inference. In the second example, BREEAM also cannot predict the $head\_of\_gov$ relation from $Modi$ to $India$, while ChatGPT makes an incorrect prediction. This fully demonstrates the superiority of our method in complex logical inference. In the third example, the rule used is different from the front two. It is a rule that satisfies the reverse constraint. We can infer the non-existence of the rule head $agency\_of\text{-}x$ by the non-existence of the rule body atom $baesd\_in0$. Both BREEAM and ChatGPT cannot satisfy reverse constraint and make an incorrect prediction, which reflects the unique advantage of our method in reverse constraints. However, BCBR is not in conflict with ChatGPT, as logic constraints can enhance the reasoning ability of large models. Thus, ChatGPT can work together with logic constraints to improve the performance of DocRE. The integration will be a interesting direction in future research.

## 5 Conclusion

In this paper, we propose a novel logic constraint framework BCBR, which utilises bidirectional

| Rule Patterns | The Mined Beta Rules With Their Beta Scores | |
|---|---|---|
| $r_{head}(e_0, e_1) \leftarrow r_1(e_0, e_1)$ | $agent\_of(e_0, e_1) \leftarrow minister\_of(e_0, e_1)$ | 0.99 |
| $r_{head}(e_0, e_2) \leftarrow r_1(e_0, e_1) \wedge r_2(e_1, e_2)$ | $in0\text{-}x(e_0, e_2) \leftarrow in0(e_0, e_1) \wedge gpe0(e_1, e_2)$ | 0.96 |
| $\neg r_{head}(e_0, e_1) \leftarrow \neg r_1(e_0, e_1)$ | $\neg head\_of(e_0, e_1) \leftarrow \neg member\_of(e_0, e_1)$ | 0.99 |
| $\neg r_{head}(e_0, e_2) \leftarrow \neg r_1(e_0, e_1) \vee \neg r_2(e_1, e_2)$ | $\neg agent\_of\text{-}x(e_0, e_2) \leftarrow \neg agency\_of(e_0, e_1) \vee \neg gpe0(e_1, e_2)$ | 0.99 |

Table 6: Case study of rules mined by our beta rule miner.

logic constraints of beta rules to regulate the output of DocRE. We are the first to propose the use of beta distribution for modeling rules, which effectively solves the problem of pseudo-rules. Then we model the reverse logic constraints and utilize bidirectional constraints of beta rules to construct rule consistency loss. By jointly training with relation classification loss, we improve the performance of DocRE. Experimental results on multiple datasets demonstrate that BCBR outperforms baseline models and other logic constraint frameworks.

## Limitations

Our BCBR brings additional rule consistency loss, resulting in a significant increase in training time. We need to traverse all rules when processing each document to generate rule consistency loss. It leads to a significant increase in time cost. We will optimize the code structure in future work to achieve convergence of the model in a relatively short period of time.

## Acknowledgements

This work was supported by National Natural Science Foundation of China (NSFC) (61972455) and the Project of Science and Technology Research and Development Plan of China Railway Corporation (N2023J044).

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

## A  Hyper-Parameter Settings

We detail the hyperparameter settings of BCBR on different datasets in Table 7.

| Hyper-param | DWIE | DocRED | Re-DocRED |
|---|---|---|---|
| maxL | 2 | 2 | 2 |
| epoch | 70 | 100 | 100 |
| $k_{sc}$ | 0.9 | 0.8 | 0.9 |
| $k_{hc}$ | 0.9 | 0.8 | 0.9 |
| $\eta$ | 0.9 | 0.9 | 0.95 |
| $\lambda$ | 1e-3 | 1e-4 | 1e-4 |

Table 7: Hyper-parameter settings on different datasets.

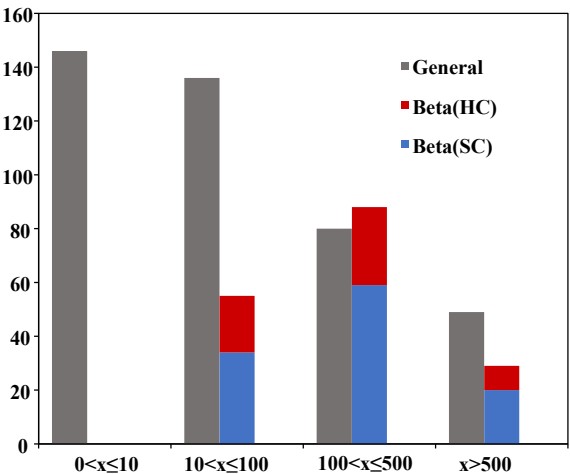

Figure 4: Comparison of the amount of rules mined by different rule miners across different intervals. The gray cluster represents the rules generated by a general rule miner, while the red and blue clusters represent the high-standard-confidenc rules and high-head-coverage rules generated by the Beta rule miner, respectively.

## B  Rule miner comparison

We analyzed the distribution of rules mined by the Beta rule miner and the general rule miner at different support intervals on DWIE dataset. Beta(SC) and Beta(HC) represent the high-standard-confidence rules and high-head-coverage rules extracted by the Beta rule extractor, respectively. The results are shown in Figure 4. We can observe that the rules mined by Beta rule miner are mostly scattered in the 101 to 500 support interval, and there are no low-support rules scattered in the 1 to 10 support interval. In contrast, the general rule miner has a large number of rules scattered in the 1 to 10 support interval. It reflects that the high quality of rules mined by the Beta rule miner is much higher than that of the general rule miner. In addition, we can observe the proportion between Beta(SC) and Beta(HC), which indicates that the rules satisfying the reverse constraint cannot be ignored. If only high-standard-confidence rules are used to constrain the relation extraction process, a large amount of consistency information among rules will be lost, leading to a decline in the performance of relation extraction.