# OpenReview forum: "Document-level Relationship Extraction by Bidirectional Constraints of Beta Rules"
_EMNLP/2023/Conference — EMNLP 2023 Main_

### Official Review · Reviewer_4D7h · 2023-08-04

**Soundness:** 4

**Excitement:**

4: Strong: This paper deepens the understanding of some phenomenon or lowers the barriers to an existing research direction.

**Paper Topic And Main Contributions:**

Certain previous studies only focused on the forward logic constraints in document-level relation extraction. In this paper, the authors present an innovative approach termed "Bidirectional Constraints of Beta Rules" (BCBR), which incorporates both forward and reverse logic constraints to assist relation extraction. BCBR reconstructs rule consistency loss by bidirectional constraints to regulate the output of the DocRE model. Extensive experiment demonstrate the superior performance of BCBR compared to the original DocRED models.

**Reasons To Accept:**

1. This paper is well-written and easy to follow.
2. The proposed BCBR outperforms both LogiRE and MILR, excelling in both relation extraction performance and logical consistency.

**Reasons To Reject:**

1. A more detailed explanation of the bidirectional logic constraints is needed. This can greatly enhance the clarity and understanding of your approach.
2.  The inclusion of additional baselines, particularly on datasets like Re-DocRED and DocRED, is advisable. A more extensive baseline comparison would contribute to a more comprehensive evaluation of your proposed approach.

**Reproducibility:**

5: Could easily reproduce the results.

**Reviewer Confidence:**

3: Pretty sure, but there's a chance I missed something. Although I have a good feel for this area in general, I did not carefully check the paper's details, e.g., the math, experimental design, or novelty.

---

> ### Author Rebuttal · Authors · 2023-08-29
>
> Q1:
> A more detailed explanation of the bidirectional logic constraints is needed. This can greatly enhance the clarity and understanding of your approach.
>
> A1:
> Thank you for your suggestion. We will add more detailed explanations for bidirectional logical rules in Section 3.3. The explanatory content is as follows:
> Bidirectional constraints encompass two aspects. Positive constraints entail the rule's body imposing constraints on the rule's head, indicating that if the rule's body exists, the rule's head must also exist. However, if the rule's body does not exist, there is no constraint on the rule's head. Conversely, reverse constraints involve the rule's head imposing constraints on the rule's body, signifying that if the rule's body does not exist, the rule's head must not exist.
>
> Q2:
> The inclusion of additional baselines, particularly on datasets like Re-DocRED and DocRED, is advisable. A more extensive baseline comparison would contribute to a more comprehensive evaluation of your proposed approach.
>
> A2:
> Thank you for your review and suggestions. We will add a new baseline, GAIN, for the DocRED dataset and Re-DocRED dataset to make the experiments more comprehensive.

---

### Official Review · Reviewer_rFoA · 2023-08-05

**Soundness:** 4

**Excitement:**

4: Strong: This paper deepens the understanding of some phenomenon or lowers the barriers to an existing research direction.

**Paper Topic And Main Contributions:**

This paper presents a new model called BCBR for DocRE. Specifically, it:
(1) mines rules utilizing beta distribution
- previous models only use confidence to mine rules
(2) combine forward and reverse logic constraints to add two additional losses to the training of the DocRE model
- previous models only consider forward rules (due to the low quality of rules)

This paper conducts experiments on DWIE, DocRED, and re-DocRED. In all three datasets, the BCBR techniques significantly improve two existing models: ATLOP and DREEAM. It also outperforms two existing techniques: LogiRE and MILR that leverage logic rules to improve DocRE.

**Questions For The Authors:**

- Just a minor suggestion, if we provide the rules mined by BR to ChatGPT, will it make better relation extraction predictions?

**Reasons To Accept:**

- The two techniques of introducing beta distribution in rule mining (BR) and training DocRE with reverse logic constraints (BC) are novel. The BC technique is quite effective.
- The paper conducts extensive experiments on both the DocRE performance and the logic consistency.

**Reasons To Reject:**

- Although the improvements over ATLOP/GAIN/DREEAM are significant. The improvement over LogiRE and MILR seems marginal on DocRED and re-DocRED, with <0.5 Test F1. Considering this paper also adopts techniques from these two papers such as forward constraint loss, the marginal improvement weakens the contribution of this paper.

- According to the ablation studies, removing BR doesn't seem to affect the performance by a lot. DREEAM (-BR) only reduces performance by 0.29/0.21 Dev Ign F1/F1. This is quite unexpected, because when the quality of rules decreases, the reverse constraint loss may also be less effective.

**Reproducibility:**

3: Could reproduce the results with some difficulty. The settings of parameters are underspecified or subjectively determined; the training/evaluation data are not widely available.

**Reviewer Confidence:**

4: Quite sure. I tried to check the important points carefully. It's unlikely, though conceivable, that I missed something that should affect my ratings.

**Typos Grammar Style And Presentation Improvements:**

- The caption of Table 3 and Table 4 should be exchanged.
- In Table 3 and Table 4, the position of "Ign F1" and "F1" should also be exchanged (F1 should be higher than Ign F1)

---

> ### Author Rebuttal · Authors · 2023-08-29
>
> Q1:
> Although the improvements over ATLOP/GAIN/DREEAM are significant. The improvement over LogiRE and MILR seems marginal on DocRED and Re-DocRED, with <0.5 Test F1. Considering this paper also adopts techniques from these two papers such as forward constraint loss, the marginal improvement weakens the contribution of this paper.
>
> A1:
> Thank you for your comments on our paper.
>
> (1)	The improvement of BCBR on the DocRED dataset and Re-DocRED dataset is indeed relatively small. Our method mitigates the issue of pseudo rules in both datasets and improves performance by complementing reverse constraints. The reasons for our minor improvements on these two datasets are as follows:
> (a) 	Our method improves by 1.54/1.57(Test Ign F1/F1) based on ATLOP and by 1.34/1.29(Test Ign F1/F1) based on DREEAM. Compared to the existing two logical methods, there is also a certain performance improvement. It proves the effectiveness of our approach.
> (b)	DWIE is a dataset containing golden rules, while DocRED and Re-DocRED comes from remote supervision, the quality of the two datasets in terms of logic is not as good as the DWIE dataset.
>
> (2)	Compared to LogiRE and MILR, our contributions mainly lie in the following two aspects:
> (a)	As far as we know, we first propose a rule miner that utilizes the Beta distribution to model rules. The regular rule miners in LogiRE and MILR may make mistakes when facing pseudo rules, but ours will not.
> (b)	We introduce reverse logic constraint to ensure that the output of DocRE models satisfies the necessity of rules. The forward logic constraint in LogiRE and MILR only solves half of the rule constraint problem.
>
> Q2:
> According to the ablation studies, removing BR doesn't seem to affect the performance by a lot. DREEAM (-BR) only reduces performance by 0.29/0.21 Dev Ign F1/F1. This is quite unexpected, because when the quality of rules decreases, the reverse constraint loss may also be less effective.
>
> A2:
> Thank you for your comments. We think the occurrence of the aforementioned situation can be attributed to the following reasons.
>
> (1)	The BR module alleviates the issue of pseudo rules and is an essential part of our framework. The DWIE dataset contains golden rules with good logical consistency, resulting in relatively fewer occurrences of pseudo rules(It still exists). As a result, the BR module has fewer problems to address in this scenario, and the contribution of the BC module becomes more significant. In DocRED dataset and Re-DocRED dataset, the contribution of the BR component would be equal to or even surpass that of BC.
>
> (2)	When using the BR module alone, as shown in Table 5 "-BC", there is an obvious significant improvement on the test dataset (Ign F1 increased from 64.82 to 66.57, F1 increased from 71.44 to 73.55). Similarly, when using the BC module alone, as shown in Table 5 "-BR", there is an improvement compared to the baseline on the test dataset (Ign F1 increased from 64.82 to 67.74, F1 increased from 71.44 to 74.42). It confirms the effectiveness of both modules from another perspective. The combined use of the BR and BC modules does not achieve the additive effect, which may be due to the consistent role of them in the rule learning process.
> Thank you for raising this question again, as it may contribute to further in-depth research on the BR module and BC module.
>
> Q3: Just a minor suggestion, if we provide the rules mined by BR to ChatGPT, will it make better relation extraction predictions?
>
> A3: We have previously conducted similar experiments, and ChatGPT cannot directly comprehend our mined rules. We need to describe them in natural language for ChatGPT to understand. However, this approach essentially treats the rules as background knowledge, lacking strong constraint effects. While it impacts the results, its influence is not as significant as using rules directly. Exploring methods to effectively constrain the output of large models using rules represents an intriguing direction and is a topic of interest for our future research work.
>
> Reply for Typos Grammar Style And Presentation Improvements:Thank you for the feedback and suggestions. We have made the corrections to the errors you mentioned and conducted another round of proofreading for the paper's format and grammar.

---

### Official Review · Reviewer_akdh · 2023-08-05

**Soundness:** 4

**Excitement:**

4: Strong: This paper deepens the understanding of some phenomenon or lowers the barriers to an existing research direction.

**Paper Topic And Main Contributions:**

The paper proposes a novel framework called "Bidirectional Constraints of Beta Rules" (BCBR) to address the challenges in Document-level Relationship Extraction (DocRE). Existing approaches using logic constraints suffer from opacity and weak logic. The BCBR framework introduces a new rule miner based on beta contribution to model rules. It then constructs forward and reverse logic constraints using these beta rules. By using bidirectional constraints, BCBR regulates the output of the DocRE model and improves relation extraction performance and logical consistency. Experimental results show that BCBR outperforms existing DocRE models by achieving higher F1 scores and logic scores.

**Reasons To Accept:**

1. The paper introduces a novel logic constraint framework, BCBR, which addresses the limitations of existing approaches in DocRE. Incorporating forward and reverse logic constraints based on beta rules is a unique and promising contribution.
2. The paper supports its claims with experimental results, demonstrating that BCBR outperforms existing DocRE models in performance and logical consistency.

**Reasons To Reject:**

1. A thorough examination of the extracted rule could be incorporated. While a comparison of the relu miners is presented in the appendix, the addition of a comprehensive case study or more detailed statistical analysis could further enrich the content.

**Reproducibility:**

3: Could reproduce the results with some difficulty. The settings of parameters are underspecified or subjectively determined; the training/evaluation data are not widely available.

**Reviewer Confidence:**

3: Pretty sure, but there's a chance I missed something. Although I have a good feel for this area in general, I did not carefully check the paper's details, e.g., the math, experimental design, or novelty.

---

> ### Author Rebuttal · Authors · 2023-08-29
>
> Q1:
> A thorough examination of the extracted rule could be incorporated. While a comparison of the rule miners is presented in the appendix, the addition of a comprehensive case study or more detailed statistical analysis could further enrich the content.
>
> A1:
> Thank you for reviewing and providing your suggestions. The focus of our research lies in rule-assisted document-level relation extraction and we conducted a detailed case study on this task in Chapter 4.5 of our paper. For the case study of rules alone, we will provide a detailed analysis of rule form cases and examples as follow:
>
> rhead(e0, e1) ← r1(e0, e1)                      |                agent_of (e0, e1) ← minister_of (e0, e1)
>
> rhead(e0, e2) ← r1(e0, e1) ∧ r2(e1, e2)           |       in0-x (e0, e2) ← in0 (e0, e1) ∧ gpe0 (e1, e2)
>
> ¬rhead(e0, e1) ← ¬r1(e0, e1)                    |             ¬head_of (e0, e1) ← ¬member_of (e0, e1)
>
> ¬rhead(e0, e2) ← ¬r1(e0, e1) ∨ ¬r2(e1, e2)      |     ¬agent_of -x (e0, e2) ← ¬agency_of (e0, e1) ∨ ¬gpe0 (e1, e2)
>
> The first two rule forms are high-confidence rules that can be used for positive constraints, while the last two rule forms are high-head coverage rules that can be used for reverse constraints. In the final submission, we will supplement it in the appendix in the form of a table.

---

### Meta-Review · Area_Chair_5kqP · 2023-09-14

**Recommendation:** 5

**Metareview:**

All reviewers agree that this is a sound and exciting paper with good results and novelty. Ablations studies should be added to improve the paper.

---

### Decision · Program_Chairs · 2023-10-07

**Decision:**

Accept-Main

**Comment:**

All reviewers agree that this is a sound and exciting paper with good results and novelty. Ablations studies should be added to improve the paper.